# Accounting for Value Changes in Cultivated Land Resources within the Karst Mountain Area of Southwest China, 2001–2020

**Lu Zhang** [1,2,†] **, Zhongfa Zhou** [3,*]**, Quan Chen** [1,†]**, Lan Wu** [4]**, Qing Feng** [3]**, Dan Luo** [4] **and Tangyin Wu** [4]

1 School of Karst Science, Guizhou Normal University, Guiyang 550001, China; sophiazl@gznu.edu.cn (L.Z.); 201407075@gznu.edu.cn (Q.C.)
2 Real Estate Registration Center of Guizhou Province, Guiyang 550001, China
3 National Engineering Research Center for Karst Rocky Desertification Control, Guiyang 550001, China; 20030170040@gznu.edu.cn
4 The State Key Laboratory Incubation Base for Karst Mountain Ecology Environment of Guizhou Province, Guiyang 550001, China; 21010170543@gznu.edu.cn (L.W.); 21010170535@gznu.edu.cn (D.L.); 21010170545@gznu.edu.cn (T.W.)
* Correspondence: fa6897@gznu.edu.cn
† These authors contributed equally to this work and should be considered co-first authors.

**Abstract:** Cultivated land resources are important natural resource assets that are related to food security and sustainable development. Due to the many restrictive factors of the karst landform on agricultural production, the quantity and quality of cultivated land in the karst mountainous areas in Southwest China are poor. Reclaiming cultivated land to develop economy or to avoid transitional reclamation to protect ecology is an important proposition in this area. Analyzing changes in the physical and monetary value of cropland resources can help us to formulate more reasonable policies for the development and utilization of cultivated land resources, and to achieve a win-win scenario for economic development and ecological protection. Using multi-source remote sensing data and 20-year landcover data obtained by the GEE platform, this paper evaluated the cropland resources of the karst mountain areas of China at the pixel level. It was found that under the apparent outflow of the physical account of the cultivated land resources, the monetary value still maintained growth, proving that the current cultivated land-use policy in Guizhou Province has significantly improved the value of local cultivated land resources.

**Keywords:** cultivated land resource; value changes; karst mountain area; remote sensing; land use policy





## 1. Introduction

Natural resource assets are important means of production that are derived from nature, and that play a decisive role in economic and social development. The coordinated relationship between resource consumption, environmental protection, and economic growth has become a subject that affects human destiny [1]. Therefore, we need to find a method for tracking changes in nature, and for determining how changes are linked to economic and other human activities, to reflect the interactions between man and nature. Considering the increasing demand for statistics on natural capital within analytical policy frameworks on environmental sustainability, human well-being, and economic growth and development, advancing this emerging statistical field has become increasingly urgent [2].

Many scholars have performed statistical accounting for various natural resources, such as land resources and forests. Natural resource asset accounting uses the theories of statistics, accounting, resource science, and other disciplines to make a reasonable valuation of natural resources within certain periods of space and time, reflecting quantitative and structural changes to their physical quantity and value [3,4]. The purpose for this is to understand the current situation of natural resources, and the reasonable occupation, use,

benefits, and disposal of natural resource assets, and finally, to solve the contradiction between resource utilization and environmental protection.

In 1993, the United Nations and the World Bank incorporated natural resources and the environment into a system of national economic accounting (SNA), and successively issued SEEA-1993 and SEEA-2003, in which physical value is used to describe interactions between the economy and the environment in various fields [5,6]. In March 2012, the 2012 System of Environmental Economic Accounting—Central Framework (SEEA-CF) was adopted as the international general guide, making it the first international statistical standard for environmental economic accounting, and it was supplemented by SEEA Experimental Ecosystem Accounting (SEEA-EEA) and SEEA Applications and Expansion. SEEA applies the accounting concepts, structures, rules, and principles of environmental information that are included in the System of National Accounts (SNA), and it uses a single framework to integrate environmental information (often measured in physical quantity) and economic information (often measured in value) [5–9]. It mainly covers the measurement of three areas: the physical flow of material and energy within and between the economy and the environment, and stocks of environmental assets and changes in these stocks, as well as environmentally related economic activities and transactions [10].

SNA, SEEA, and SEEA-EEA account research provides a good theoretical basis for the accounting of natural resource assets, but traditional SNA and SEEA accounting takes the natural environment as a kind of production material and adopts methods for which it can be presented to reflect the stock of the means of production and the flow in economic activities. Experience exists in related areas of assessment, such as land-cover and land-use statistics, but the integration of different areas of expertise into an accounting framework is new. In the latest SEEA-EEA specification, the principle of using surveying and mapping results has also been emphasized. At present, many studies also focus on how to use remote sensing data to support natural capital accounting [11]. Since natural resources have inherent location attributes, natural resources of the same quantity or quality will show great geographical differentiation in different locations; that is, simple presentation and accounting methods will not include the important spatial characteristics of natural resources. As a result, using multi-remote sensing data to conduct natural resource value not only allows the quantity and quality indicators of accounting objects to be obtained quickly, reducing the workload of manual investigation, but it can also evaluate the accounting results in the spatial dimension, so that the accounting results can better serve the decision-making processes.

In October 2016, UNSD, UNEP, CBD, and EU initiated NCAVES. The project lasted 3 years and was implemented in China, Brazil, India, Mexico, and South Africa. This project aimed to assist China in advancing the country's knowledge agenda for environmental and ecosystem accounting, and to initiate the pilot testing of SEEA Experimental Ecosystem Accounting (SEEA-EEA), as well as ecosystem valuation and macro-economic analysis, with a view toward improving the management of natural biotic resources, ecosystems, and their services at the national level, and mainstreaming biodiversity and ecosystems in national level policy planning and implementation [12]. Guizhou Province is one of the pilots in China, and many scholars have conducted much research into the natural resource balance sheet, GEP, ESV, and other fields, but the subject, object, and method of accounting need to be unified [13–15].

Cropland accounts for 10.20% of the global land surface area, which is the most important resource for agricultural production, and it plays an important role in ensuring food security, ecological security, and sustainable development [16,17]. The cultivated land resource is a natural resource that has been domesticated by human beings. Its growth and decline are not only restricted by natural laws, but are significantly affected by human activities. Compared to other kinds of natural resource assets, cultivated land resources can not only provide necessary food for survival, but they also participate in the energy transformation and material cycle of nature as an ecosystem, which is closely related to human society. This thus establishes how cropland value contributes to physical and

monetary changes in long time series, which can assist with the analysis of the change range, flow characteristics, and reasons for change.

The formation of a karst landform is the result of the long-term dissolution of limestone and other soluble rocks by groundwater or surface water. The surface water is dissolved and eroded along the joints and fissures of soluble rocks, forming an uneven and broken surface shape. As one of the three karst-concentrated distribution areas in the world, the karst area in southern China has many factors that are not conducive to agricultural production. These factors, such as bedrock exposure, small soil stock, and discontinuous distribution [18,19], make agricultural planting difficult, and the cost of cultivated land management is very high. Additionally, due to the development of karst, the surface water is difficult to maintain, which means there is a serious water shortage in this region, but at the same time, the discharge of surface water in the rainy season is too late, causing water accumulation in some karst depressions. Therefore, karst areas in southern China are often accompanied by poverty; because both the quality and quantity of cultivated land are poor, the more cultivated the land is, the poorer the people, and the contradiction between man and land is very prominent. As the core area of karst in southern China, Guizhou has serious rocky desertification and a large area of rock exposure. By exploring the impact of human activities on cultivated land, we can determine the positive policies that can improve the value of cultivated land resources, something that is of great significance for ameliorating the current situation of poverty in China's poor areas within the karst [20–22].

Landcover data provide the most direct feedback when accounting for cultivated land resource physical quantity, but cropland resource assessment methods will inevitably require more detailed spatial data. As the development of remote sensing and big-data technology have already brought a new approach towards accounting, we can obtain multi-source remote sensing data more quickly to assist with the accounting work, improve the accuracy of the accounting, and reduce the cost. This research aimed to realize the dynamic monitoring of the spatial pattern evolution of cropland resources via physical accounting, using multi-remote sensing data [23]. It can make up for the defects in the SEEA-CF accounting framework, which only presents data rather than spatial information. Meanwhile, in order to quantify the change rules of the cultivated land resource value, and to observe whether effective land management policies have been adopted, this paper evaluated the changes of cultivated land resource value in Guizhou Province from 2001 to 2020. By analyzing the impact of the economy and other human activities on cropland, it proved that the current cultivated land use policy in Guizhou has significantly improved the value of local cultivated land resources. This provides a reference for the rational utilization of cultivated land resources.

## 2. Materials and Methods

### 2.1. Study Area

Guizhou province is located in the inland area of Southwest China, to the east of Yunnan Guizhou Plateau, and is located between 24°37′–29°13′ N and 103°36′–190°35′ E, which is an important ecological barrier in the upper reaches of the Yangtze River and the Pearl River [24,25]. Meanwhile, as the junction of the Eurasian plate and the Indian Ocean plate, its terrain is high in the west and low in the east, tilting from the middle to the north, and from the east and to the south. The landform of the whole province can be divided into four basic types: plateau, mountain, hill, and basin. Moreover, Guizhou province is one of the three karst-concentrated distribution areas in the world, the core area of East Asia, which is also the largest distribution area and the strongest conical karst development in China. With high mountains, deep valleys, and steep terrain, 92.5% of the area of the province is mountainous and hilly, and 109,100 square km comprises exposed karst landform, which means the surface is extremely fragmented and lacks the cropland resources for agriculture [26–28]. In addition, due to the increasing population, the cultivated land area continues to reduce, meaning that the percapita cultivated land area is less than 300 square meters, which is far lower than the average level in China [29].

Moreover, the proportion of cultivated land with a thick soil layer, high fertility, and good conditions of water conservation is low (Figure 1).

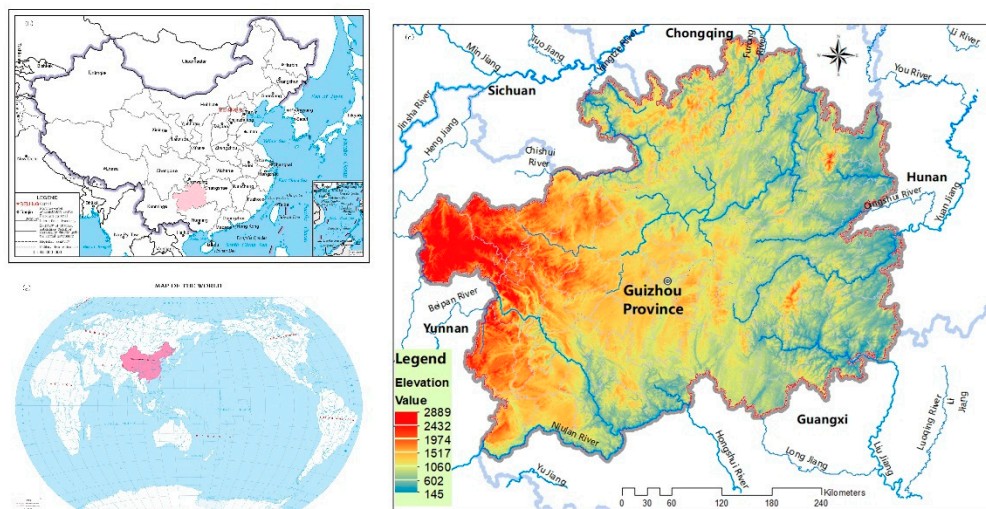

**Figure 1.** Location of the study area.

*2.2. Dataset*

2.2.1. Spatial Data

We used Google Earth Engine (GEE) to gather and to calculate the spatial data for this analysis. GEE is an interactive platform that provides geospatial processing services that are powered by the Google Cloud Platform [30]. With Earth Engine, we can perform geospatial processing at a scale that is free of charge, and we can carry out high-impact, data-driven scientific research involving large geospatial datasets [31,32]. In this research, we adopted multi-remote sensing time series data from 2000 to 2020, to detect the impact of land use changes on the value of cropland resources. Landcover data were derived from images collected by the MODIS sensor (the MCD12Q1 V6 product), which provides global land cover types at yearly intervals (250 m × 250 m). The digital elevation models (DEMs) used Shuttle Radar Topography Mission (SRTM) data at a 30 m resolution. Additionally, we estimated the Landsat net primary production (NPP) using Landsat Surface Reflectance for CONUS (Landsat net primary production CONUS) [33]. Beyond these, we selected the GPM data (Monthly Global Precipitation Measurement v6) to revise the existing results of ecological value. Global Precipitation Measurement (GPM) is an international satellite mission that provides next-generation observations of rain and snow worldwide, every three hours. The Integrated Multi-Satellite Retrievals for GPM (IMERG) is a unified algorithm that provides rainfall estimates by combining data from all passive-microwave instruments in the GPM Constellation.

2.2.2. Socioeconomic Data

Socioeconomic data, including the yields of major farm crops (YMFC), the gross output value of farming (GOVF), the gross domestic product (GDP), the permanent resident population (PRP) and the employment in agriculture were obtained from the Guizhou statistical yearbook (2001–2021) (http://stjj.guizhou.gov.cn/ accessed on 4 April 2022). In addition, we derived the grain prices from 2001 to 2020 from the "The National Compilation of Cost-benefit data of Agricultural Products" as a reference (Table 1).

**Table 1.** Data sources for assessing cultivated land resources value.

| | Resource Type | Data Sources |
|---|---|---|
| Spatial Data | Land cover (MCD12Q1 V6)<br>Digital elevation models (DEMs)<br>Landsat net primary production (NPP)<br>Global precipitation measurement (GPM) | Google Earth Engine Plateform<br>(https://developers.google.cn/earth-engine/<br>datasets) accessed on 4 April 2022 |
| Socioeconomic Data | Yields of major farm crops (YMFC)<br>Gross output value of farming (GOVF)<br>Gross domestic product (GDP)<br>Permanent resident population (PRP)<br>Financial expenditure<br>Employments in agriculture | Guizhou statistical yearbook (2001–2021)<br>(http://stjj.guizhou.gov.cn/) accessed on<br>4 April 2022 |
| | Grain prices | The National Compilation of Cost-benefit data<br>of Agricultural Products |

## 2.3. Methods

### 2.3.1. Cropland Resources Value Accounting Framework

To make a scientific evaluation of the value of cropland resources, we established three accounting accounts [34]: the physical quantity account, the conditional account, and the monetary account [35–38]. Among them, the physical quantity account was used to reflect the changes in the number and scope of cultivated land in the study area from 2000 to 2020, and to provide necessary data for value accounting, while the quality account was used to record the quality status of cultivated land in the study area. Since it is obvious that the value of cropland varies along the quality status, there will be significant differences in crop yield and ecological function. Finally, the monetary account includes two parts. One is the direct value, also called the use value or the commodity value, which is the value that is formed by people's direct harvesting, which is the output value of agricultural products provided by cropland resources. This part can be calculated by the market price method, because agricultural products can directly enter circulation as commodities [39]. The other part is the indirect value, which refers to the ecological service ability of cropland resources as a part of the natural environment when they exist in a natural way, as well as the value of natural resource assets that are used to meet human spiritual, cultural, and moral needs, and social development [40] (Table 2).

**Table 2.** Indicators for assessing cultivated land resources value.

| Account | First-Level Indicators | Second-Level Indicators |
|---|---|---|
| Physical Account | Extent | Area |
| | Biomass provision | Crop Production |
| Conditional Account | Site conditions | Elevation<br>Slope |
| | Landscape index | Patch Density (PD)<br>Edge Density (ED)<br>Area-Weighted Mean Shape Index (AWMSI)<br>Fragmentation Index of Patch Numbers (FN)<br>Fragmentation Shape Index (FS)<br>Aggregation Index (AI) |
| Monetary Account | Direct value | Crop Market Value |
| | Indirect value | Gas Regulation<br>Climate Regulation<br>Environmental Purification<br>Hydrological Regulation<br>Soil Conservation<br>Maintenance of Nutrient Cycles<br>Biodiversity<br>Aesthetic Landscape |

The direct value is the gross output value of various grains, tubers, oil crops, vegetables, and other crops in Guizhou Province. The indirect value is the sum of the value equivalent for each ecological function. The annual cropland resources value is the direct value plus the indirect value.

$$V_T = V_D + V_{ID} \tag{1}$$

$$V_T = \sum_{i=1}^{n} V_{Di} \tag{2}$$

$$V_{ID} = \sum_{i=1}^{n} V_{IDi} \tag{3}$$

In the formula, $V_T$ is the total monetary value of cropland resources, $V_D$ is the gross output value, and $V_{ID}$ is the total value equivalent of cropland ecological function.

2.3.2. Landscape Index

Cropland fragmentation refers to the fragmentation, dispersion, and size of the cultivated land due to natural or human factors, and the area of each cultivated land is relatively small, showing a decentralized and disorderly pattern, which is a long-term dynamic process [41–43]. For the karst mountain areas, the high mountains and deep valleys lead to obvious cutting terrain, and cultivated land can only be distributed on gentle slopes or small flat land. Therefore, the degree of cultivated land fragmentation is a very typical quality evaluation index in karst areas and plays a decisive role in the realization of the value of cultivated land resources [44,45].

Research on the impact of cultivated land fragmentation on the landscape scale of cultivated land can directly reflect changes in cultivated land fragmentation. In this study, we used the open-source Python library to compute landscape metrics, and the following six indicators were selected to measure the cultivated land landscape [46]:

- Patch Density (PD)

This indicator refers to the number of cultivated land patches per unit area in the study area, and it has an important impact on biological protection, material, and energy distribution. This index reflects the situation in which the concentrated and contiguous cultivated land is divided into small patches, which directly reflects upon the connotation of cultivated land landscape fragmentation [47].

$$PD = n/A \tag{4}$$

$n$ is the number of the patches; $A$ is the total area.

- Edge Density (ED)

This is an index that is used to analyze the shape of land patches, revealing the degree of cropland segmentation, as well as being a direct reflection of the degree of cultivated land fragmentation. The greater the edge density, the higher the degree of cultivated land division, and the more scattered the layout [48].

$$ED = P/A \tag{5}$$

$P$ is the total perimeter of all cropland patches; $A$ is the total area.

- Fragmentation Index of Patch Numbers (FN)

The patch size is the most basic spatial feature and it directly affects the mechanization level of agricultural production. As such, this index is used to measure the degree of fragmentation of the landscapes.

$$FN = (N-1)/MPS \tag{6}$$

$MPS$ is the mean patch size; $N$ is the number of cropland patches.

- Area-Weighted Mean Shape Index (AWMSI)

Since an irregular shape leads to a reduction in the actual planting area within the total area, the farming production cost per unit area will be increased. However, with an increase in the patch size, the impact caused by the irregular shape will gradually weaken. Considering this phenomenon, AWMSI is taken to be one of the indicators to measure the degree of the cultivated landscape.

$$\text{AMWSI} = \sum_{i=1}^{n}\left[\left(\frac{0.25P_i}{\sqrt{a_i}}\right)(a_i/A)\right] \tag{7}$$

$n$ is the number of cropland patches; $P_i$ is the perimeter of the patches; $a_i$ is the area of the patches; $A$ is the total area of cropland.

- Fragmentation Shape Index (FS)

This index is used to reflect the internal combination of cultivated land patches. The distribution of cultivated land patches becomes more scattered as the index increases. Additionally, the internal combination simultaneously becomes more complex.

$$\text{FS} = 1 - 1/MSI \tag{8}$$

$$MSI = \sum_{i=1}^{n}(0.25P_i/\sqrt{a_i})/N \tag{9}$$

$MSI$ is the mean shape index; $a_i$ is the patch area; $P_i$ is the perimeter of the patch; $N$ is the number of cropland patches.

- Aggregation Index (AI).

This index reflects the degree of patch agglomeration within the landscape type. When the value is larger, the landscape is composed of a few large patches, and when the value is smaller, the landscape is composed of many small patches [49].

$$\text{AI} = \frac{e_i}{\text{max\_}e_i} \times 100 \tag{10}$$

$$\text{max\_}e_i = \begin{cases} 2n(n-1), & m = 0 \\ 2n(n-1) + 2m - 1, & m \le n \\ 2n(n-1) + 2m - 2, & m > n \end{cases}, \quad \left(m = A_i - n^2\right) \tag{11}$$

$e_i$ is the number of edges that the patches have in common; $\text{max\_}e_i$ is the maximum number of edges that the patches have in common; $P_i$ is the perimeter of the patch; $n$ is the edge length of the largest integer square that does not exceed the total area of the cropland area.

### 2.3.3. Revisions of the Ecological Value Equivalent Factors

Costanza et al. proposed the principle and method of ecosystem service value estimation [50], but their methods were criticized because they resulted in the ecological value of the cultivated land being significantly low. Therefore, Chinese researchers such as Xie Gaodi revised Costanza's assessment framework based on China's economic situation, land use, and vegetation types, and developed an assessment method for China's ecosystem service value based on the unit area value equivalence factor [51–53] (Appendix A). As the ecological function value consequently varies with the internal structure and external form of ecosystem, constantly changing within different regions or different periods, we conducted two revisions to obtain the final ecosystem service value equivalent of Guizhou [54]:

1. Previous studies have shown that the ecosystem function is positively correlated with NPP and precipitation. As such, we used two temporal and spatial factors (NPP and precipitation) to modify the ecosystem service value equivalent table of China for each year.

$$F_i = \begin{cases} P_i \times F_{n1} \\ R_i \times F_{n2} \end{cases} \tag{12}$$

$F_i$ refers to the unit area value equivalent of the ecological service function for each year; $P_i$ refers to the NPP regulation factor; $R_i$ refers to the precipitation regulation factor; $F_{n1}$ represents the value equivalent per unit area of China for gas regulation, climate regulation, environmental purification, nutrient conservation, and biodiversity maintenance; and $F_{n2}$ represents the value equivalent of China's unit area of hydrological regulation function.

2.  According to Costanza's research, the economic value of ecological service value equivalent factors is 54 USD/hm$^2$ (1997). Combined with China's grain production income, Chinese scholars have calculated that the economic value of an ecological service value equivalent factor in China is 449 CNY/hm$^2$ (58.5 USD/hm$^2$ in 2007), using the shadow land rent method. However, the price index and grain yield vary interannually, and so to reflect the indirect value change of cultivated land resources more accurately, we revised the economic value by year to form the final economic value of the ecological function, to make it suitable for the study area [55].

$$E_{Vi} = \frac{1}{7} \sum_{i=1}^{n} \frac{m_i p_i q_i}{M} \tag{13}$$

$E_{Vi}$ refers to the economic value of an ecological service value for equivalent factors of cropland resources in each year; $m_i$ refers to the area of crops; $p_i$ refers to the average price of crops; $q_i$ represents the output of agricultural products; $n$ represents the types of crop products.

## 3. Results

### 3.1. Physical Account Changes

#### 3.1.1. Spatial Changes of Guizhou Province

Through the analysis of the land cover data of the study area from 2001 to 2020, it was found that the cropland resources in Guizhou Province experienced a small increase from 2001 to 2003, and they have then decreased year-by-year since 2004 (Figure 2). By 2020, the cropland resources had reduced to 3768.34 km$^2$, which means that the number had decreased by 55.52% compared to 2001. At the same time, it is easy to see that the cultivated land resources in Guizhou Province are very scarce. The proportion of cultivated land resources only accounted for 5.35% at the highest level (2003), while this figure reduced to 2.14% in 2020 (Table 3). Moreover, with the increase in the population, the percapita cultivated land resources in Guizhou Province show absolute scarcity, from 223.04 m$^2$ in 2001 to 97.68 m$^2$ in 2020.

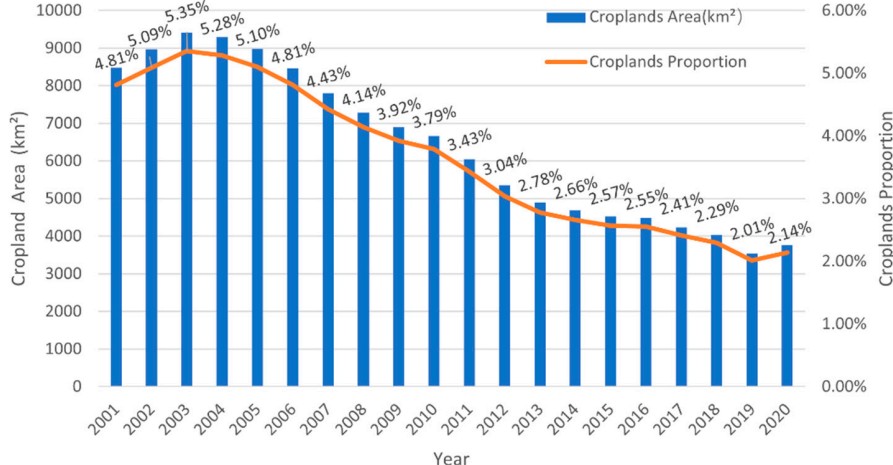

**Figure 2.** Cropland resources area of Guizhou Province.

**Table 3.** Area changes of cropland resources.

| | 2001 | 2002 | 2003 | 2004 | 2005 | 2006 | 2007 | 2008 | 2009 | 2010 |
|---|---|---|---|---|---|---|---|---|---|---|
| Cropland Area (km$^2$) | 8473.22 | 8968.61 | 9416.08 | 9294.78 | 8980.94 | 8469.23 | 7800.17 | 7286.11 | 6902.27 | 6667.85 |
| Croplands Proportion | 4.81% | 5.09% | 5.35% | 5.28% | 5.10% | 4.81% | 4.43% | 4.14% | 3.92% | 3.79% |
| Croplands per capita (m$^2$) | 223.04 | 233.74 | 243.31 | 238.08 | 240.78 | 229.52 | 214.76 | 202.62 | 195.14 | 191.66 |
| | **2011** | **2012** | **2013** | **2014** | **2015** | **2016** | **2017** | **2018** | **2019** | **2020** |
| Croplands Area (km$^2$) | 6047.527 | 5356.4 | 4894.82 | 4683.90 | 4526.89 | 4494.66 | 4236.63 | 4032.42 | 3543.72 | 3768.34 |
| Croplands Proportion | 3.43% | 3.04% | 2.78% | 2.66% | 2.57% | 2.55% | 2.41% | 2.29% | 2.01% | 2.14% |
| Croplands per capita (m$^2$) | 174.33 | 153.74 | 139.77 | 133.52 | 128.24 | 119.60 | 111.40 | 105.51 | 92.09 | 97.68 |

Secondly, each patch of land cover data was calculated, and the time series changes of each pixel was analyzed for different years, with the finding that the area of cropland resources experienced both transfer-in and transfer-out in the same year. During this period, the positive area changes in the cultivated land area are in a "U" shape, while the negative area changes are represented by a wave form. Moreover, with the transfer proportions of −9.43% (−798.82 km$^2$), −9.28% (−786.03 km$^2$), and −8.55% (−724.78 km$^2$), 2007, 2012 and 2019 became troughs. The overall distribution of the total change area was similar to that of the negative area (Figure 3).

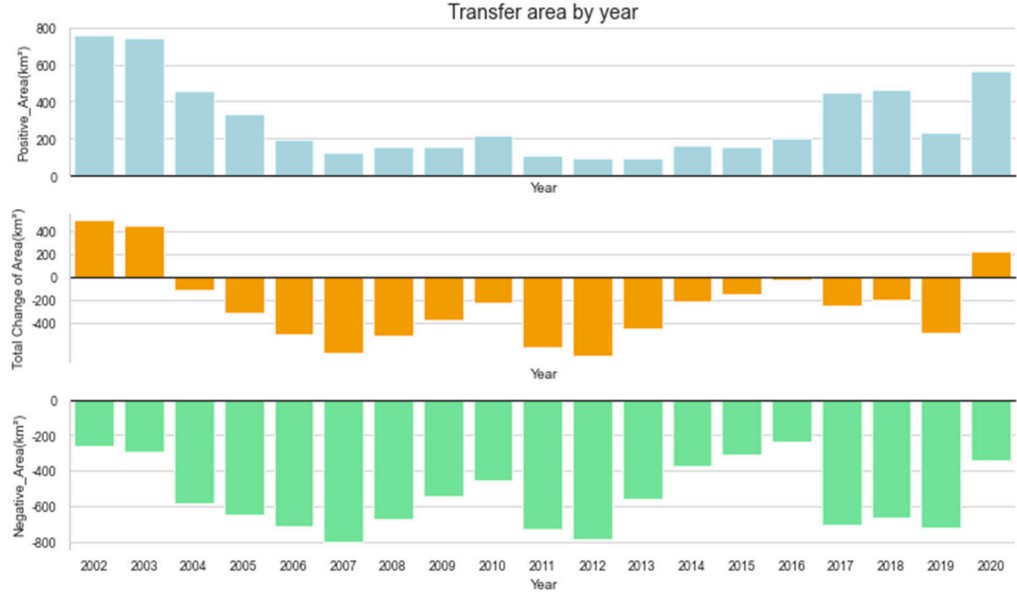

**Figure 3.** Cropland resources transfer area of Guizhou Province.

According to the International Geosphere-Biosphere Programme (IGBP) classification (Appendix A), there are 15 types of land cover in Guizhou Province. As can be seen from Figure 4, the cultivated land resources in Guizhou Province are mainly transferred with grasslands, savannas, and cropland/natural vegetation mosaics. Recent land cover data over the last 20 years show that the transfer of cropland/natural vegetation mosaics account for an average of 66% of the total transfer area. The type of cropland/natural vegetation mosaics are mosaics of small-scale cultivation, with 40–60% of natural tree, shrub, or herbaceous vegetation in a pixel. The increase in cropland/natural vegetation mosaics shows that the fragmentation of cultivated land resources in Guizhou Province increased from 2014 to 2019 (Table 4).

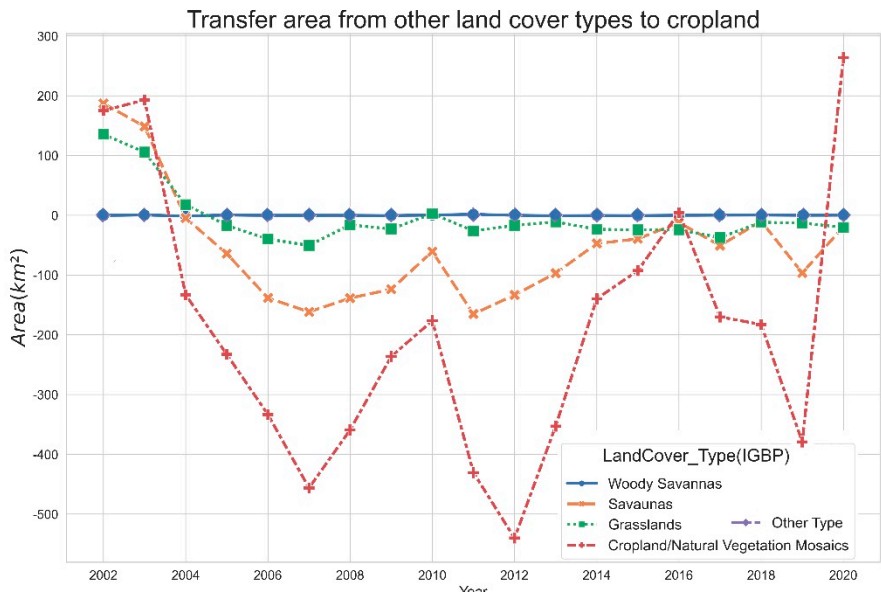

**Figure 4.** Transfer area of each landcover type in Guizhou Province.

**Table 4.** Transfer area and proportion of main landcover types.

| | | Woody Savannas | Savannas | Grasslands | Cropland/Natural Vegetation Mosaics |
|---|---|---|---|---|---|
| 2002 | Area (km$^2$) | −0.32 | 187.22 | 135.42 | 175.06 |
| | Proportion | 0.06% | 37.79% | 27.34% | 35.34% |
| 2003 | Area (km$^2$) | 0.45 | 148.45 | 105.53 | 193.05 |
| | Proportion | 0.10% | 33.17% | 23.58% | 43.14% |
| 2004 | Area (km$^2$) | −1.23 | −4.68 | 17.78 | −133.17 |
| | Proportion | 0.79% | 2.98% | 11.34% | 84.90% |
| 2005 | Area (km$^2$) | 0.34 | −64.31 | −16.98 | −232.89 |
| | Proportion | 0.11% | 20.47% | 5.40% | 74.13% |
| 2006 | Area (km$^2$) | −0.22 | −138.36 | −39.78 | −333.35 |
| | Proportion | 0.04% | 27.04% | 7.77% | 65.14% |
| 2007 | Area (km$^2$) | −0.22 | −161.69 | −50.45 | −456.26 |
| | Proportion | 0.03% | 24.17% | 7.54% | 68.19% |
| 2008 | Area (km$^2$) | −0.22 | −138.62 | −16.06 | −359.17 |
| | Proportion | 0.04% | 26.97% | 3.12% | 69.87% |
| 2009 | Area (km$^2$) | −0.56 | −123.74 | −23.01 | −236.31 |
| | Proportion | 0.15% | 32.24% | 6.00% | 61.56% |
| 2010 | Area (km$^2$) | 0.38 | −60.77 | 2.67 | −176.47 |
| | Proportion | 0.16% | 25.27% | 1.11% | 73.37% |
| 2011 | Area (km$^2$) | 1.63 | −165.25 | −26.13 | −430.57 |
| | Proportion | 0.26% | 26.50% | 4.19% | 69.05% |
| 2012 | Area (km$^2$) | −0.11 | −133.43 | −17.06 | −540.31 |
| | Proportion | 0.02% | 19.31% | 2.47% | 78.18% |
| 2013 | Area (km$^2$) | −0.76 | −96.93 | −11.00 | −352.89 |
| | Proportion | 0.17% | 21.00% | 2.38% | 76.45% |
| 2015 | Area (km$^2$) | −0.11 | −12.31 | −24.25 | 4.43 |
| | Proportion | 0.35% | 25.13% | 15.53% | 58.99% |
| 2016 | Area (km$^2$) | −0.11 | −12.31 | −24.25 | 4.43 |
| | Proportion | 0.26% | 29.95% | 59.01% | 10.78% |
| 2017 | Area (km$^2$) | 0.16 | −50.65 | −37.40 | −170.15 |
| | Proportion | 0.06% | 19.60% | 14.48% | 65.86% |
| 2018 | Area (km$^2$) | 0.28 | −10.01 | −11.73 | −182.74 |
| | Proportion | 0.13% | 4.89% | 5.73% | 89.25% |
| 2019 | Area (km$^2$) | 0.00 | −96.45 | −12.99 | −379.26 |
| | Proportion | 0.00% | 19.74% | 2.66% | 77.61% |
| 2020 | Area (km$^2$) | 0.12 | −19.17 | −20.16 | 263.83 |
| | Proportion | 0.04% | 6.32% | 6.65% | 86.99% |

3.1.2. Crop Production Changes in Guizhou Province

According to the research presented, there are two dimensions of changes in the output of agricultural products in Guizhou Province. The first is the change in quantity. The total output of agricultural products increased from 12.67 million tons up to 42.34 million tons, from 2001 to 2020, with an increase rate of 234.19%. The second is that the planting structure changed greatly, which is reflected in the changes in the crop types within the same crop type, and the quantitative changes between the different types.

The main agricultural products in Guizhou Province can be divided into grains, potatoes, oil crops, and others, of which the output of vegetables far exceeds other products, reaching 29.9087 million tons in 2020. Rice and corn are the main grain, showing little interannual change and fluctuating in the range of 605,940. Tubers increased slightly; Irish potatoes are the main crop and showed obviously changes. The median output from 2001 to 2020 was 1.535 million tons, and the third quarter was 2.335 million tons. Rapeseed is the main oil plant crop, accounting for more than 70%. As the economy has continued to develop, ramie has been completely replaced by other types of crops (Figure 5).

The proportion of grain compared to the total agriculture products of Guizhou Province increased from 72.55% in 2001 and plummeted to 16.34% in 2020; at the same time, the output of oil crops and tubers has also decreased by nearly half in 20 years, while other high-value-added crops that increased from 916.1 thousand tons (7.23%) in 2001 surged to 30.73 million tons (72.58%) in 2020 (Table 5).

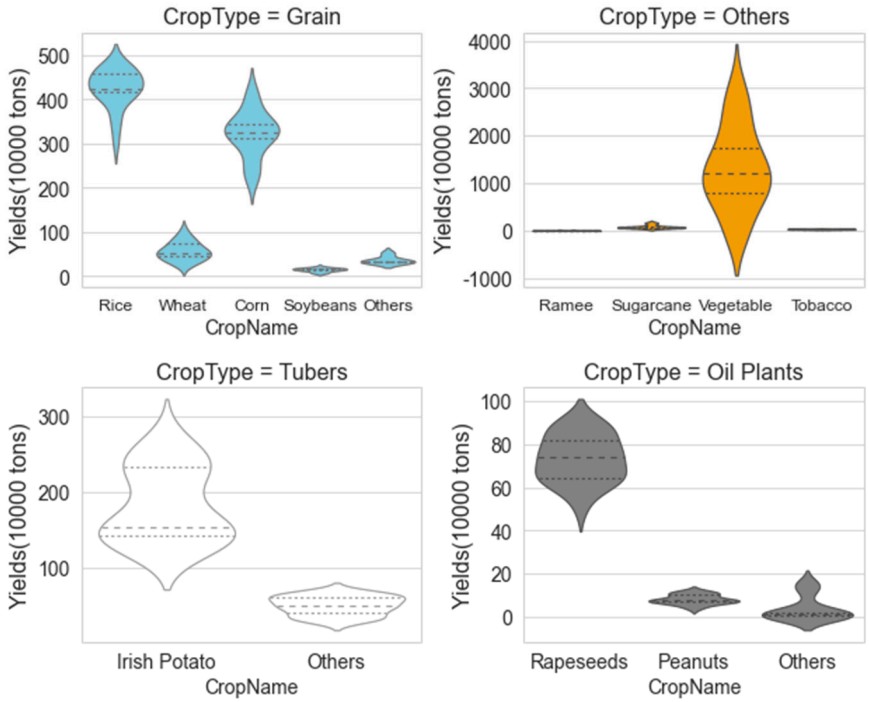

**Figure 5.** Crop production distribution in Guizhou Province (2001–2020).

**Table 5.** Crops production of Guizhou Province (10,000 tons).

| | Grain | | Oil Plants | | Others | | Tubers | | Total Yields |
|---|---|---|---|---|---|---|---|---|---|
| 2001 | 919.2 | 72.55% | 71.32 | 5.63% | 91.61 | 7.23% | 184.9 | 14.59% | 1267.03 |
| 2002 | 829.7 | 68.49% | 72.48 | 5.98% | 104.73 | 8.65% | 204.5 | 16.88% | 1211.41 |
| 2003 | 903.5 | 45.33% | 72.31 | 3.63% | 816.72 | 40.97% | 200.8 | 10.07% | 1993.33 |
| 2004 | 939.32 | 44.69% | 82.71 | 3.94% | 869.38 | 41.37% | 210.26 | 10.00% | 2101.67 |
| 2005 | 906.24 | 41.59% | 84.89 | 3.90% | 942.08 | 43.23% | 245.82 | 11.28% | 2179.03 |
| 2006 | 820.07 | 41.23% | 68.24 | 3.43% | 882.59 | 44.38% | 217.93 | 10.96% | 1988.83 |
| 2007 | 869.73 | 40.53% | 69.66 | 3.25% | 975.53 | 45.46% | 231.13 | 10.77% | 2146.05 |

**Table 5.** *Cont.*

| | Grain | | Oil Plants | | Others | | Tubers | | Total Yields |
|---|---|---|---|---|---|---|---|---|---|
| 2008 | 911.67 | 39.18% | 68.39 | 2.94% | 1100.62 | 47.30% | 246.33 | 10.59% | 2327.01 |
| 2009 | 918.76 | 37.87% | 78.68 | 3.24% | 1179 | 48.60% | 249.51 | 10.29% | 2425.95 |
| 2010 | 901.9 | 36.60% | 60.34 | 2.45% | 1291.3 | 52.41% | 210.4 | 8.54% | 2463.94 |
| 2011 | 605.13 | 26.52% | 78.85 | 3.46% | 1326.15 | 58.12% | 271.77 | 11.91% | 2281.9 |
| 2012 | 804.91 | 29.73% | 87.38 | 3.23% | 1540.9 | 56.91% | 274.59 | 10.14% | 2707.78 |
| 2013 | 718.88 | 25.46% | 91.53 | 3.24% | 1701.53 | 60.27% | 311.11 | 11.02% | 2823.05 |
| 2014 | 790.33 | 25.78% | 98.05 | 3.20% | 1829.23 | 59.67% | 348.17 | 11.36% | 3065.78 |
| 2015 | 815.89 | 25.48% | 101.34 | 3.16% | 1920.9 | 59.99% | 364.11 | 11.37% | 3202.24 |
| 2016 | 828.38 | 24.09% | 113.66 | 3.31% | 2132.24 | 62.01% | 364 | 10.59% | 3438.28 |
| 2017 | 808.94 | 22.11% | 109.82 | 3.00% | 2370.18 | 64.78% | 369.6 | 10.10% | 3658.54 |
| 2018 | 732.59 | 18.93% | 112.62 | 2.91% | 2698.65 | 69.72% | 327.11 | 8.45% | 3870.97 |
| 2019 | 707.57 | 17.81% | 103.01 | 2.59% | 2819.22 | 70.95% | 343.67 | 8.65% | 3973.47 |
| 2020 | 692.04 | 16.34% | 103.4 | 2.44% | 3073.27 | 72.58% | 365.59 | 8.63% | 4234.3 |

*3.2. Conditional Account Changes*

3.2.1. Changes in Site Conditions

We used the GEE to calculate the DEM data for the cultivated land resources in Guizhou Province, which showed that the cultivated land resources in Guizhou Province are mainly distributed near the elevations of 1320 m and 2220 m. The mean elevation increased by approximately 130 m from 2001 to 2020, but the standard deviation decreased significantly, implying that the elevation of the cultivated land resources in Guizhou Province is gradually concentrated to the average value (Table 6). Therefore, it can also be judged that the elevation of the cultivated land resources in Guizhou Province have shown an overall increase.

**Table 6.** Statistical results of the elevation of the cultivated land resources in Guizhou Province.

| | Mean | Median | Std-Dev | Mix | Max |
|---|---|---|---|---|---|
| 2001 | 1577.95 | 1461 | 526.76 | 299 | 2831 |
| 2002 | 1557.5 | 1440 | 524.91 | 229 | 2831 |
| 2003 | 1540.92 | 1426 | 527.19 | 229 | 2831 |
| 2004 | 1529.13 | 1415 | 534.46 | 229 | 2831 |
| 2005 | 1516.96 | 1403 | 538.91 | 229 | 2831 |
| 2006 | 1521.53 | 1402 | 536.46 | 229 | 2834 |
| 2007 | 1535.18 | 1407 | 531.34 | 229 | 2815 |
| 2008 | 1549.44 | 1418 | 527.26 | 229 | 2815 |
| 2009 | 1571.46 | 1433 | 524.21 | 229 | 2815 |
| 2010 | 1587.78 | 1448 | 517.44 | 229 | 2834 |
| 2011 | 1623.42 | 1483 | 516.99 | 229 | 2834 |
| 2012 | 1648.67 | 1516 | 520.09 | 229 | 2834 |
| 2013 | 1674.58 | 1564 | 516.2 | 229 | 2834 |
| 2014 | 1682.55 | 1576 | 510.25 | 261 | 2834 |
| 2015 | 1689.86 | 1585 | 506.38 | 261 | 2834 |
| 2016 | 1682.65 | 1559 | 500.84 | 260 | 2834 |
| 2017 | 1659.46 | 1508 | 493.31 | 260 | 2811 |
| 2018 | 1661.55 | 1520 | 494.06 | 260 | 2811 |
| 2019 | 1700.18 | 1633 | 492.13 | 241 | 2769 |
| 2020 | 1693.96 | 1587 | 475.21 | 262 | 2757 |

Meanwhile, through the statistics of the slope of each cultivated land pixel, it was found that 70–80% of the cultivated land resources in Guizhou Province are distributed in areas with a slope of less than 10°. With the evolution of the distribution pattern of the cultivated land resources, the changes in the slope of the cultivated land resources can be divided into two stages (Figure 6). First, from 2001 to 2004, the number of cultivated land pixels with a slope of less than 25° continued to increase. In 2004, 46812 pixels were distributed in areas below 5°, accounting for 50.68% of the total area of cultivated land resources. Additionally, there were 28556 pixels with a slope of between 5° and 10°,

accounting for 29.75% of the total area of cultivated land resources in that year. Secondly, from 2005 to 2020, the area of cultivated land with a slope of more than 25° decreased significantly, with a maximum change rate of more than 80%, and the cultivated land area with a slope above 40° completely disappeared (Table 7).

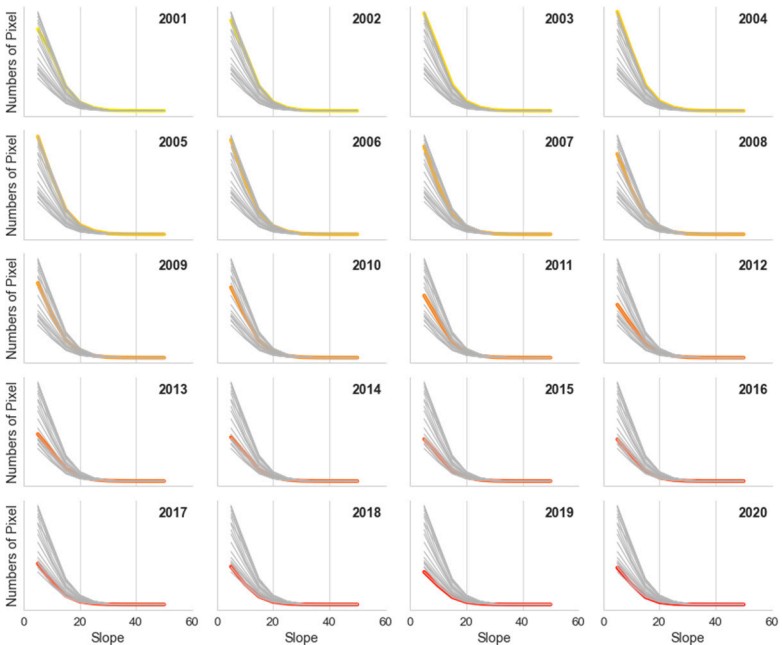

**Figure 6.** Slope changes of cropland resources in Guizhou Province.

**Table 7.** Pixels counts of the slopes of cropland resources in Guizhou Province.

| Year | 0–5° | 5°–10° | 10°–15° | 15°–20° | 20°–25° | 25°–30° | 30°–35° | 35°–40° | 40°–45° | 45°–50° |
|---|---|---|---|---|---|---|---|---|---|---|
| 2001 | 38,599 | 26,799 | 11,162 | 3935 | 1391 | 324 | 86 | 32 | 8 | 2 |
| 2002 | 42,765 | 27,879 | 11,562 | 4118 | 1422 | 338 | 91 | 34 | 8 | 2 |
| 2003 | 46,059 | 29,252 | 12,208 | 4396 | 1533 | 373 | 106 | 38 | 11 | 2 |
| 2004 | 46,812 | 28,556 | 11,993 | 4438 | 1553 | 363 | 94 | 31 | 10 | 2 |
| 2005 | 46,240 | 27,146 | 11,490 | 4369 | 1540 | 340 | 84 | 21 | 6 | 2 |
| 2006 | 44,636 | 25,236 | 10,535 | 3907 | 1335 | 286 | 74 | 19 | 6 | 2 |
| 2007 | 41,474 | 22,768 | 9360 | 3399 | 1127 | 256 | 64 | 17 | 6 | 2 |
| 2008 | 38,024 | 21,051 | 8620 | 3068 | 1024 | 233 | 59 | 16 | 6 | 2 |
| 2009 | 35,303 | 20,253 | 8232 | 2857 | 948 | 220 | 55 | 16 | 6 | 2 |
| 2010 | 33,253 | 19,613 | 7985 | 2790 | 936 | 218 | 57 | 16 | 6 | 2 |
| 2011 | 29,369 | 17,972 | 7294 | 2559 | 868 | 209 | 57 | 16 | 6 | 2 |
| 2012 | 25,028 | 15,550 | 6414 | 2260 | 781 | 187 | 48 | 15 | 8 | 2 |
| 2013 | 22,197 | 14,107 | 5833 | 2103 | 720 | 181 | 49 | 12 | 8 | 1 |
| 2014 | 20,752 | 13,127 | 5398 | 1925 | 636 | 147 | 38 | 10 | 5 | 1 |
| 2015 | 19,857 | 12,350 | 5021 | 1783 | 563 | 133 | 32 | 8 | 4 | 1 |
| 2016 | 19,717 | 11,945 | 4768 | 1597 | 496 | 117 | 32 | 7 | 4 | 1 |
| 2017 | 19,346 | 10,659 | 4098 | 1380 | 463 | 85 | 31 | 7 | 4 | 1 |
| 2018 | 17,932 | 9610 | 3763 | 1242 | 406 | 72 | 28 | 2 | 3 | 0 |
| 2019 | 15,385 | 8921 | 3498 | 1162 | 368 | 68 | 20 | 2 | 0 | 0 |
| 2020 | 17,405 | 9697 | 3497 | 1056 | 312 | 54 | 19 | 3 | 0 | 0 |
| Max Change rate | 62.82% | 66.85% | 71.35% | 76.20% | 79.89% | 85.11% | 82.08% | 92.11% | 100.00% | 100.00% |

### 3.2.2. Landscape Index Changes

By calculating the six dimensions of the landscape index for the cultivated land resources in Guizhou Province from 2001 to 2020 (Table 8), we found that the patch density (PD), edge density (ED), and aggregation index (AI) in Guizhou Province increased first, and then decreased. Meanwhile, the area-weighted mean shape index (AWMSI) showed a negative trend by year. Additionally, the change trend in the fragmentation index of the patch numbers (FN) was negatively correlated with the fragmentation shape index (FS). According to the calculation results, the PD decreased from 0.0117 in 2001 to 0.007 in

2020, indicating that the fragmentation of cultivated land resources in Guizhou Province improved, and the ED decreased from 0.9973 in 2001 to 0.5548 in 2020, indicating that the shape of the cultivated land gradually became more regular [47]. The AWMSI decreased from 7.954 in 2001 to 6.001 in 2020, indicating that the distribution of cultivated land plots tends to be centralized. It can be seen that the vulnerability of cultivated land in Guizhou Province has been reduced.

**Table 8.** Landscape index changes of cultivated land resources in Guizhou Province.

| Year | PD | ED | FN | AWMSI | FS | AI |
|------|------|------|------|------|------|------|
| 2001 | 0.0117 | 0.9973 | 14.802622 | 7.954 | 0.28310273 | 79.5801 |
| 2002 | 0.012 | 1.0353 | 14.666524 | 7.9734 | 0.28356498 | 79.9096 |
| 2003 | 0.0119 | 1.0653 | 13.813862 | 7.6421 | 0.29263634 | 80.2707 |
| 2004 | 0.0113 | 1.0362 | 12.507077 | 7.6585 | 0.29358576 | 80.6063 |
| 2005 | 0.0106 | 0.9917 | 11.415985 | 7.6782 | 0.29088073 | 80.8496 |
| 2006 | 0.0097 | 0.9384 | 10.262608 | 7.3696 | 0.29567545 | 80.878 |
| 2007 | 0.0093 | 0.8818 | 10.142569 | 6.8898 | 0.29473165 | 80.5931 |
| 2008 | 0.0089 | 0.846 | 9.9882939 | 6.5778 | 0.29903267 | 80.132 |
| 2009 | 0.0088 | 0.8114 | 10.413669 | 6.5818 | 0.29173454 | 79.9631 |
| 2010 | 0.0089 | 0.7977 | 10.807041 | 6.3748 | 0.2917847 | 79.6394 |
| 2011 | 0.0084 | 0.7385 | 10.757928 | 6.4825 | 0.28861066 | 79.3885 |
| 2012 | 0.0081 | 0.6836 | 11.313965 | 6.597 | 0.28310273 | 78.6491 |
| 2013 | 0.0078 | 0.6439 | 11.563548 | 6.4575 | 0.27917538 | 78.1715 |
| 2014 | 0.0081 | 0.638 | 12.935312 | 6.6323 | 0.27436325 | 77.4089 |
| 2015 | 0.0083 | 0.6324 | 13.99703 | 6.729 | 0.26691592 | 76.8763 |
| 2016 | 0.0085 | 0.6407 | 14.919812 | 6.7727 | 0.26524614 | 76.3702 |
| 2017 | 0.0081 | 0.6163 | 14.078087 | 6.3173 | 0.27028605 | 76.0021 |
| 2018 | 0.0083 | 0.6109 | 15.651382 | 6.0193 | 0.26975318 | 75.0247 |
| 2019 | 0.0071 | 0.54 | 13.104289 | 5.8462 | 0.27103076 | 75.385 |
| 2020 | 0.007 | 0.5548 | 11.886296 | 6.0001 | 0.274942 | 76.1123 |

PD: patch density. ED: edge density. FN: patch numbers. AI: aggregation index. AWMSI: area-weighted mean shape index. FS: fragmentation shape index.

### 3.3. Monetary Account Changes

From 2001 to 2020, with the development of the economy, the direct economic value of cultivated land resources in Guizhou Province increased rapidly. The production value increased from CNY 27,995 million per year to CNY 180,025 million per year, with an increase of 543%. It could be evidenced (Table 9) that in the past 20 years, with the adjustment of the industrial structure, the main labor force flowed to the secondary and tertiary industries with a high added value and high income, reducing the number of agricultural employees in Guizhou. In 2001, there were 1.36 million agricultural employments, while in 2020, only 0.634 million people were employed in agriculture. At the same time, the per capita output value increased from CNY 2046.42/y to CNY 28,395.11/y, an increase of 12.87-fold. However, according to the calculation of the price index of agricultural products in "The National Compilation of Cost-benefit data of Agricultural Products", the sales price of agricultural products in China only increased by 181.96% from 2001 to 2020. In other words, the direct economic value of cultivated land resources in Guizhou Province still improved significantly after removing the influence of the interannual differences in prices.

**Table 9.** Direct economic value of cropland resources in Guizhou Province.

| Year | Cross Output Value (Million Yuan) | Agriculture Employment ($10^4$) | Cross Output Value per Capita (CNY) |
|------|------|------|------|
| 2001 | 27,995 | 1368 | 2046.42 |
| 2002 | 27,888 | 1354 | 2059.68 |
| 2003 | 46,672 | 1322 | 3530.41 |

**Table 9.** *Cont.*

| Year | Cross Output Value (Million Yuan) | Agriculture Employment ($10^4$) | Cross Output Value per Capita (CNY) |
|------|------|------|------|
| 2004 | 52,464 | 1288 | 4073.29 |
| 2005 | 33,353 | 1268 | 2630.36 |
| 2006 | 34,797 | 1247 | 2790.46 |
| 2007 | 39,220 | 1388 | 2825.65 |
| 2008 | 30,848 | 1350 | 2285.04 |
| 2009 | 33,050 | 1299 | 2544.26 |
| 2010 | 38,561 | 1210 | 3186.86 |
| 2011 | 43,084 | 1194 | 3608.38 |
| 2012 | 56,132 | 1189 | 4720.94 |
| 2013 | 64,612 | 1180 | 5475.59 |
| 2014 | 85,189 | 1171 | 7274.89 |
| 2015 | 109,654 | 1162 | 9436.66 |
| 2016 | 119,650 | 883 | 13,550.35 |
| 2017 | 130,643 | 828 | 15,778.11 |
| 2018 | 143,929 | 765 | 18,814.19 |
| 2019 | 156,647 | 700 | 22,378.14 |
| 2020 | 180,025 | 634 | 28,395.11 |

As for the indirect value, according to the revised ecological value per unit area of farmland ecosystems and the equivalent of the ecological service value per unit area in Guizhou Province (Table 10), we calculated the indirect value of cultivated land resources (Table 11). Under the dual influence of cultivated land resource area falling and the grain price index increasing, the indirect economic value of cultivated land resources in Guizhou Province first increased, and then decreased. Among them, it reached a peak of CNY 7775.25 million in 2009, but the overall decrease was no more than 3%, indicating that the ecological function of the cultivated land resources in Guizhou Province is still well-protected while the economy is developing (Table 12).

**Table 10.** Ecosystem service equivalent value per unit area of cropland ecosystem in Guizhou.

| | Regulating Services | | | | Supporting Services | | Cultural Services | |
|---|---|---|---|---|---|---|---|---|
| | Gas Regulation | Climate Regulation | Environmental Purification | Hydrological Regulation | Soil Conservation | Maintenance of Nutrient Cycle | Biodiversity | Aesthetic Landscape |
| 2001 | 3.25 | 1.70 | 0.49 | 5.27 | 1.90 | 0.57 | 0.62 | 0.27 |
| 2002 | 3.01 | 1.57 | 0.46 | 5.36 | 1.76 | 0.52 | 0.57 | 0.25 |
| 2003 | 2.77 | 1.45 | 0.42 | 4.64 | 1.62 | 0.48 | 0.53 | 0.23 |
| 2004 | 2.70 | 1.41 | 0.41 | 5.40 | 1.58 | 0.47 | 0.52 | 0.23 |
| 2005 | 2.91 | 1.52 | 0.44 | 4.44 | 1.70 | 0.51 | 0.56 | 0.25 |
| 2006 | 2.98 | 1.56 | 0.45 | 5.00 | 1.74 | 0.52 | 0.57 | 0.25 |
| 2007 | 3.09 | 1.61 | 0.47 | 5.47 | 1.80 | 0.54 | 0.59 | 0.26 |
| 2008 | 3.00 | 1.57 | 0.46 | 5.51 | 1.75 | 0.52 | 0.57 | 0.25 |
| 2009 | 3.03 | 1.58 | 0.46 | 4.49 | 1.77 | 0.53 | 0.58 | 0.26 |
| 2010 | 2.72 | 1.42 | 0.41 | 4.63 | 1.59 | 0.47 | 0.52 | 0.23 |
| 2011 | 2.84 | 1.48 | 0.43 | 4.17 | 1.66 | 0.49 | 0.54 | 0.24 |
| 2012 | 2.86 | 1.50 | 0.43 | 4.56 | 1.67 | 0.50 | 0.55 | 0.24 |
| 2013 | 3.15 | 1.64 | 0.48 | 4.16 | 1.84 | 0.55 | 0.60 | 0.27 |
| 2014 | 2.95 | 1.54 | 0.45 | 5.99 | 1.72 | 0.51 | 0.56 | 0.25 |
| 2015 | 3.03 | 1.58 | 0.46 | 5.48 | 1.77 | 0.53 | 0.58 | 0.26 |
| 2016 | 3.08 | 1.61 | 0.47 | 4.84 | 1.80 | 0.54 | 0.59 | 0.26 |
| 2017 | 3.07 | 1.60 | 0.47 | 4.97 | 1.79 | 0.53 | 0.59 | 0.26 |
| 2018 | 2.75 | 1.44 | 0.42 | 4.81 | 1.61 | 0.48 | 0.53 | 0.23 |
| 2019 | 3.19 | 1.66 | 0.48 | 5.39 | 1.86 | 0.55 | 0.61 | 0.27 |
| 2020 | 2.86 | 1.49 | 0.43 | 5.83 | 1.67 | 0.50 | 0.55 | 0.24 |

**Table 11.** Indirect value of cropland resources in Guizhou Province (million CNY).

| | Regulating Services | | | | Supporting Services | | | Cultural Services |
|---|---|---|---|---|---|---|---|---|
| | Gas Regulation | Climate Regulation | Environmental Purification | Hydrological Regulation | Soil Conservation | Maintenance of Nutrient Cycle | Biodiversity | Aesthetic Landscape |
| 2001 | 1042.31 | 544.58 | 158.10 | 1687.38 | 608.99 | 181.53 | 199.09 | 87.83 |
| 2002 | 1037.58 | 542.11 | 157.39 | 1850.00 | 606.23 | 180.70 | 198.19 | 87.44 |
| 2003 | 1185.91 | 619.61 | 179.89 | 1989.32 | 692.89 | 206.54 | 226.52 | 99.94 |
| 2004 | 1353.88 | 707.36 | 205.36 | 2708.69 | 791.03 | 235.79 | 258.61 | 114.09 |
| 2005 | 1337.41 | 698.76 | 202.87 | 2036.25 | 781.41 | 232.92 | 255.46 | 112.70 |
| 2006 | 1443.82 | 754.36 | 219.01 | 2424.08 | 843.58 | 251.45 | 275.79 | 121.67 |
| 2007 | 1735.44 | 906.72 | 263.24 | 3075.90 | 1013.97 | 302.24 | 331.49 | 146.25 |
| 2008 | 1587.42 | 829.38 | 240.79 | 2913.23 | 927.48 | 276.46 | 303.22 | 133.77 |
| 2009 | 1699.16 | 887.76 | 257.74 | 2512.78 | 992.77 | 295.92 | 324.56 | 143.19 |
| 2010 | 1560.85 | 815.50 | 236.76 | 2655.25 | 911.96 | 271.83 | 298.14 | 131.53 |
| 2011 | 1012.34 | 528.92 | 153.56 | 1486.50 | 591.48 | 176.31 | 193.37 | 85.31 |
| 2012 | 1321.58 | 690.49 | 200.46 | 2102.56 | 772.16 | 230.16 | 252.44 | 111.37 |
| 2013 | 1121.78 | 586.10 | 170.16 | 1483.78 | 655.42 | 195.37 | 214.27 | 94.53 |
| 2014 | 1069.32 | 558.69 | 162.20 | 2173.93 | 624.77 | 186.23 | 204.25 | 90.11 |
| 2015 | 908.11 | 474.46 | 137.75 | 1645.79 | 530.58 | 158.15 | 173.46 | 76.53 |
| 2016 | 788.70 | 412.07 | 119.63 | 1239.54 | 460.81 | 137.36 | 150.65 | 66.46 |
| 2017 | 841.59 | 439.71 | 127.66 | 1364.21 | 491.72 | 146.57 | 160.75 | 70.92 |
| 2018 | 661.68 | 345.71 | 100.37 | 1157.39 | 386.60 | 115.24 | 126.39 | 55.76 |
| 2019 | 698.34 | 364.86 | 105.93 | 1181.51 | 408.02 | 121.62 | 133.39 | 58.85 |
| 2020 | 924.71 | 483.14 | 140.27 | 1885.77 | 540.28 | 161.05 | 176.63 | 77.93 |

**Table 12.** Changes of economic value of cropland resources in Guizhou Province (million CNY).

| Year | Direct Value | Indirect Value | Total Value |
|---|---|---|---|
| 2001 | 27,995.00 | 4509.80 | 32,504.80 |
| 2002 | 27,888.00 | 4659.64 | 32,547.64 |
| 2003 | 46,672.00 | 5200.60 | 51,872.60 |
| 2004 | 52,464.00 | 6374.81 | 58,838.81 |
| 2005 | 33,353.00 | 5657.76 | 39,010.76 |
| 2006 | 34,797.00 | 6333.76 | 41,130.76 |
| 2007 | 39,220.00 | 7775.25 | 46,995.25 |
| 2008 | 30,848.00 | 7211.75 | 38,059.75 |
| 2009 | 33,050.00 | 7113.88 | 40,163.88 |
| 2010 | 38,561.00 | 6881.84 | 45,442.84 |
| 2011 | 43,084.00 | 4227.79 | 47,311.79 |
| 2012 | 56,132.00 | 5681.21 | 61,813.21 |
| 2013 | 64,612.00 | 4521.42 | 69,133.42 |
| 2014 | 85,189.00 | 5069.51 | 90,258.51 |
| 2015 | 109,654.00 | 4104.83 | 113,758.83 |
| 2016 | 119,649.56 | 3375.23 | 123,024.78 |
| 2017 | 130,642.73 | 3643.13 | 134,285.85 |
| 2018 | 143,928.56 | 2949.13 | 146,877.69 |
| 2019 | 156,647.00 | 3072.52 | 159,719.52 |
| 2020 | 180,025.00 | 4389.76 | 184,414.76 |

## 4. Discussion

### 4.1. Analysis of Reasons for the Change in Physical and Conditional Account

Based on the results of this paper, the cultivated land resources in Guizhou Province declined continually after a short increase, and the reduced area was mainly transformed into natural vegetation and grassland, especially as the steep slope terraces disappeared from 2001 to 2020 [56]. These changes are closely related to the continuous implementation of the policy mandating the return of farmland to forest and grassland land types in Guizhou Province [57,58]. In particular, the intensity of returning farmland to forest and grassland in poverty-stricken areas of Guizhou has increased during a critical period of poverty alleviation, such as farmland with a slope of more than 25 degrees, severely sandy farmland, sloped farmland of 15–25 degrees in areas with important water sources,

and steep sloped terraces, the conversion of all of which are examples of remarkable achievements [44,59]. At the same time, it is worth mentioning that on the premise of the obvious outflow of the physical account of the cultivated land resources, the output of agricultural products has still shown a huge increase [60,61]. It is not difficult to find that the output of grain crops decreased, but that the output of high value-added crops such as tobacco and vegetables increased. On the one hand, as people's quality of life improves, people's eating habits tend to become more diversified and healthier, leading to an increase in the demand for more value-added commodities in human society. Namely, the relationship between supply and demand in the market has guided farmers to the crop types that they choose to grow. On the other hand, in order to get rid of poverty in Guizhou Province, the government has implemented relevant policies with regard to the adjustment of the agricultural planting structure to improve the income of farmers in karst mountainous areas [62].

Meanwhile, six dimensions of cropland landscape indicators, such as edge density and the area-weighted means shape index were used as a measure of cultivated land fragmentation, to analyze the landscape change of cultivated land resources in Guizhou Province from 2001 to 2020 [63]. These indicators have decreased significantly, indicating that the fragmentation of cultivated land resources has been alleviated through land consolidation and ecological restoration projects [64].

### 4.2. Analysis of Reasons for the Change in Monetary Account

As for the results of the analysis on the monetary account, the monetary value of cultivated land resources in Guizhou Province has increased greatly over the past 20 years. It is interesting that, with the obvious outflow of the physical quantity account of cultivated land resources, the growth rate of the monetary value of agricultural products is still significant. More importantly, the settlement of the issue cannot be achieved by simply expanding the cultivated area or by increasing the employed population. It can be seen from the above data that agricultural employment and the cultivated land area in Guizhou Province have decreased by more than 50%, but the value of the agricultural products created per capita has increased by 12-fold. The improvement of cropland quality and the development of technology have led to a rise in cropland resource value in Guizhou Province. While the direct value has increased, the indirect value has not fallen sharply, indicating that the ecological environment has been protected during economic development [65].

In addition, we collected government expenditure data from the study area over the past 20 years (Appendix B). In 2001, the local government spent CNY 4.25 billion on farming, forestry, and water conservation, which has increased to CNY 10.431 billion in 2020, and this investment had increased 23-fold. This included the giving of subsidies to encourage farmers to adjust their planting structure, increasing the construction of water conservation facilities to ensure irrigation conditions, and conducting corresponding education on agricultural technology to improve farmers' planting skills. These policies ensure that the adjustment of planting structure can be quickly completed within the study area [66,67].

At the same time, with high mountains and steep slopes, the construction cost of roads and bridges is very high, which makes the transportation and sales of agricultural products inconvenient. The local government increased its investment in transportation and other infrastructure from CNY 4.31 billion in 2001 to CNY 34.15 billion in 2020, realizing the County-to-County Expressway and the "village to village" hardened road in Guizhou Province. From the results of this study, it seems that reasonable policy guidance and sustained high-level financial investment have led to a significant increase in the value of cultivated land resources in this area [68,69].

Affected by karst landforms, Guizhou Province has serious soil erosion, serious rocky desertification, and a lack of cultivated land resources [70]. In order to improve rocky desertification and soil erosion, Guizhou Province has conducted large-scale rocky desertification prevention and control projects. Meanwhile, the financial investment for environmental

protection has increased from CNY 2.667 billion in 2006 to CNY 14.615 billion in 2020. By accounting for the cultivated land resource assets in Guizhou Province over the past 20 years, it has been found that even under relatively bad natural conditions, the asset value of cultivated land resources can apparently be improved and realized in a win-win situation of economic development and ecological protection, through the guidance of reasonable land use methods and scientific land policies [71].

### 4.3. Shortcomings/Uncertainties of This Research

However, the landcover data selected is of 250 m resolution in this study. For the karst mountainous areas, some sloped croplands of small areas may not have been identified, or they could have been identified as cropland/natural vegetation mosaics, which may lead to deviations in the evaluation results. In addition, only site conditions and landscape indexes are selected for conditional accounting. For cropland resources, soil quality, soil physical and chemical properties, and obstacle factors are also important measures. In future research, multiple measures should be added to the conditional account, so as to more comprehensively develop knowledge regarding the quality changes in cultivated land resources.

### 5. Conclusions

In this paper, multi-remote sensing data were used to calculate the physical and conditional account changes of the cultivated land resources in Guizhou Province at the pixel level, which may make up for the deficiency of traditional accounting of natural capital by presentation. At the same time, according to the characteristics of karst landforms in the study area, landscape factors were added to the conditional account, which will assist us with precisely analyzing the reasons for the change of monetary account. Through this research, we drew the following conclusions.

1.  In the physical account, the cultivated land resources in Guizhou Province showed an obvious downward trend, but the planting structure of agricultural products showed obvious changes, and the gross output increased significantly. This shows that the value of the cultivated land is not strongly related to the size of the land area.
2.  In the condition account, the quality of the cultivated land resources in Guizhou Province improved. Specifically, the fragmentation of the cultivated land improved, and the area of cultivated land on steep slopes decreased. This shows that the local governance policy on cultivated land is effective.
3.  In the monetary account, the monetary value of the cropland resources in Guizhou Province increased greatly and rapidly. Additionally, an increase in economic value did not place negative impacts upon the ecological value of the cultivated land. This shows that reasonable policy and financial investment are of positive significance for the sustainable utilization of the cultivated land resources.

Based on the above conclusions, we believe that it is very necessary to introduce additional representative factors into the accounting of cultivated land resource value in the study area. Evaluation and research into the value of cultivated land resources in the karst mountainous areas in Southwest China can provide a good reference for scholars of related fields. Moreover, in this case, reasonable policies, such as returning farmland to forest and adjusting agricultural planting structure have very positive impacts on the value of cultivated land resources and the improvement of farmers' benefits in this area. This is not only an evaluation of the effect of land policy implementation through quantitative methods, but it is also is a useful demonstration for leaders in other areas with similar difficulties; an active exploration of the sustainable utilization of cultivated land resources.

There are still many deficiencies in this study, such as the low accuracy of land use classification, the factor of the condition account being imperfect, and so on. This is the direction in which we will continue to study in the future. It is hoped that a more perfect and universal accounting framework that is suitable for karst areas can be developed in the future, so that the evaluation results can better guide sustainable land use in the study area.

**Author Contributions:** Conceptualization, Z.Z. and L.Z.; methodology, Q.C.; software, Q.F. and L.Z.; formal analysis, L.Z. and Q.C.; investigation, L.W., D.L., and T.W.; data curation, Q.F.; writing—original draft preparation, L.Z.; writing—review and editing, Z.Z., L.Z., and Q.C.; visualization, L.Z.; supervision, Q.C.; project administration, L.Z.; funding acquisition, Z.Z. All authors have read and agreed to the published version of the manuscript.

**Funding:** This research was funded by the NSFC regional project, "Research on the coupling mechanism between ecological assets and regional poverty in karst rocky desertification areas (41661088)", by "Guizhou Province's high-level innovative talent training plan 'hundred' level talents (Qiankehe platform talents [2016] 5674)" and a special study of Guizhou Provincial Department of natural resources on the "Construction of evaluation system of real estate economic operation system in Guizhou Province" (520000215RSUFG5DLMENO).

**Institutional Review Board Statement:** Not applicable.

**Informed Consent Statement:** Not applicable.

**Data Availability Statement:** Not applicable.

**Conflicts of Interest:** The authors declare no conflict of interest.

## Appendix A

**Table A1.** MCD12Q1 International Geosphere-Biosphere Programme (IGBP) legend and class descriptions.

| Name | Value | Description |
|---|---|---|
| Evergreen Needleleaf Forests | 1 | Dominated by evergreen conifer trees (canopy > 2 m). Tree cover > 60%. |
| Evergreen Broadleaf Forests | 2 | Dominated by evergreen broadleaf and palmate trees (canopy > 2 m). Tree cover > 60%. |
| Deciduous Needleleaf Forests | 3 | Dominated by deciduous needleleaf (larch) trees (canopy > 2 m). Tree cover > 60%. |
| Deciduous Broadleaf Forests | 4 | Dominated by deciduous broadleaf trees (canopy > 2 m). Tree cover > 60%. |
| Mixed Forests | 5 | Dominated by neither deciduous nor evergreen (40–60% of each) tree type (canopy > 2 m). Tree cover > 60%. |
| Closed Shrublands | 6 | Dominated by woody perennials (1–2 m height), > 60% cover. |
| Open Shrublands | 7 | Dominated by woody perennials (1–2 m height), 10–60% cover. |
| Woody Savannas | 8 | Tree cover 30–60% (canopy > 2 m). |
| Savannas | 9 | Tree cover 10–30% (canopy > 2 m). |
| Grasslands | 10 | Dominated by herbaceous annuals (<2 m) |
| Permanent Wetlands | 11 | Permanently inundated lands with 30–60% water cover and >10% vegetated cover. |
| Croplands | 12 | At least 60% of area is cultivated cropland. |
| Urban and Built-up Lands | 13 | At least 30% impervious surface area, including building materials, asphalt, and vehicles. |
| Cropland/Natural Vegetation Mosaics | 14 | Mosaics of small-scale cultivation, 40–60% with natural trees, shrubs, or herbaceous vegetation. |
| Permanent Snow and Ice | 15 | At least 60% of area is covered by snow and ice for at least 10 months of the year. |
| Barren | 16 | At least 60% of area is non-vegetated barren (sand, rock, soil) areas with less than 10% vegetation. |
| Water Bodies | 17 | At least 60% of area is covered by permanent water bodies. Unclassified 255 Has not received a map label because of missing inputs. |

**Table A2.** Ecosystem service equivalent value per unit area.

| Ecosystem Classification | | Provisioning Services | | | Regulating Services | | | | Supporting Services | | | Cultural Services |
|---|---|---|---|---|---|---|---|---|---|---|---|---|
| Primary Classification | Secondary Classification | Food Production | Raw Material Production | Water Supply | Gas Regulation | Climate Regulation | Environmental Purification | Hydrological Regulation | Soil Conservation | Maintenance of Nutrient Cycles | Biodiversity | Aesthetic Landscape |
| Crop land | Dryland | 0.85 | 0.4 | 0.02 | 0.67 | 0.36 | 0.1 | 0.27 | 1.03 | 0.12 | 0.13 | 0.06 |
| | Paddy field | 1.36 | 0.09 | −2.63 | 1.11 | 0.57 | 0.17 | 2.72 | 0.01 | 0.19 | 0.21 | 0.09 |
| Forest | Coniferous | 0.22 | 0.52 | 0.27 | 1.7 | 5.07 | 1.49 | 3.34 | 2.06 | 0.16 | 1.88 | 0.82 |
| | Mixed coniferous | 0.31 | 0.71 | 0.37 | 2.35 | 7.03 | 1.99 | 3.51 | 2.86 | 0.22 | 2.6 | 1.14 |
| | Broad-leaved | 0.29 | 0.66 | 0.34 | 2.17 | 6.5 | 1.93 | 4.74 | 2.65 | 0.2 | 2.41 | 1.06 |
| | Shrub | 0.19 | 0.43 | 0.22 | 1.41 | 4.23 | 1.28 | 3.35 | 1.72 | 0.13 | 1.57 | 0.69 |
| Grassland | Grass | 0.1 | 0.14 | 0.08 | 0.51 | 1.34 | 0.44 | 0.98 | 0.62 | 0.05 | 0.56 | 0.25 |
| | Scrub | 0.38 | 0.56 | 0.31 | 1.97 | 5.21 | 1.72 | 3.82 | 2.4 | 0.18 | 2.18 | 0.96 |
| | Meadow | 0.22 | 0.33 | 0.18 | 1.14 | 3.02 | 1 | 2.21 | 1.39 | 0.11 | 1.27 | 0.56 |
| Wetland | Wetlands | 0.51 | 0.5 | 2.59 | 1.9 | 3.6 | 3.6 | 24.23 | 2.31 | 0.18 | 7.87 | 4.73 |
| Desert | Desert | 0.01 | 0.03 | 0.02 | 0.11 | 0.1 | 0.31 | 0.21 | 0.13 | 0.01 | 0.12 | 0.05 |
| | Bare ground | 0 | 0 | 0 | 0.02 | 0 | 0.1 | 0.03 | 0.02 | 0 | 0.02 | 0.01 |
| Waters | Water system | 0.8 | 0.23 | 8.29 | 0.77 | 2.29 | 5.55 | 102.24 | 0.93 | 0.07 | 2.55 | 1.89 |
| | Glacial snow | 0 | 0 | 2.16 | 0.18 | 0.54 | 0.16 | 7.13 | 0 | 0 | 0.01 | 0 |

**Appendix B**

**Table A3.** Statistics of financial expenditure in Guizhou Province (section) (unit: CNY billion).

| Unit: Billion CNY | General Public Budget Expenditure | Farming, Forestry and Water Conservancy | Transportation | Energy Saving and Environment Protection |
|---|---|---|---|---|
| 2001 | 27.52 | 4.25 | 4.31 | - |
| 2002 | 31.67 | 4.86 | 3.64 | - |
| 2003 | 33.24 | 4.53 | 3.16 | - |
| 2004 | 41.84 | 7.23 | 3.83 | - |
| 2005 | 52.07 | 7.66 | 4.12 | - |
| 2006 | 61.041 | 6.155 | 4.193 | - |
| 2007 | 79.54 | 8.75 | 4.88 | 2.67 |
| 2008 | 105.54 | 12.17 | 4.94 | 4.04 |
| 2009 | 137.23 | 20.41 | 12.08 | 5.53 |
| 2010 | 163.15 | 24.68 | 10.96 | 5.43 |
| 2011 | 224.94 | 27.85 | 30.52 | 5.55 |
| 2012 | 275.57 | 36.19 | 28.86 | 6.57 |
| 2013 | 308.266 | 40.031 | 29.979 | 6.644 |
| 2014 | 354.28 | 44.719 | 43.201 | 8.534 |
| 2015 | 393.95 | 53.426 | 39.225 | 9.649 |
| 2016 | 426.236 | 62.938 | 28.997 | 12.709 |
| 2017 | 461.252 | 61.205 | 33.691 | 12.539 |
| 2018 | 502.968 | 66.484 | 38.149 | 13.438 |
| 2019 | 594.874 | 99.89 | 34.779 | 18.853 |
| 2020 | 573.95 | 102.431 | 34.15 | 14.615 |

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
