# Peer review of "Accounting for Value Changes in Cultivated Land Resources within the Karst Mountain Area of Southwest China, 2001–2020"

_land, doi:10.3390/land11060765_

Round 1
Reviewer 1 Report
Line 105: new paragraph - "The formation of karst landform is the result of long-term..."
Line 108-120: This sentence is too long and complicated to understand. Rephrase. Make shorter sentences. Correct English phrasing.
Line 133-138: Here should be clearly stated aim/objectives of the study, not the results. Add the aim.
Line 180-186: I would expect here a table or link to the appendix presenting major data mentioned in the text.
Line 205: table 1 - abbreviation in the table must be explained below the table
Line 309: Add additional subchapter - 2.4 Uncertainties and shortcomings. Explain what was missing/not available so that results would be even better. Which data would improve the presentation results and assessment of the area.
Line 322-324: Figure 2- year axis remove a thousand separators for the year. Add title on secondary Y-axis is missing
Line 325: Table 2 - remove a thousand separators for the year.
Line 334: Figure 3: There is no need for different colours of bars between years. They have no added value.
Line 336: Figure 4 - is not referenced in the text. I assume the year 2001 is the base year. Mark that on the figure
Line 349: Figure 5 - the symbol for Evergreen Forest and cropland are practically similar. Change the symbol for cropland to be more distinct.
Figure 2-8 and Table 2-6: FIgure and tables are sometimes presenting the same data - double presentation. Please consider moving the unneeded figures or tables to the appendix. Please decide what will you keep in the main text, figure or the table.
Line 352-356: It would be interesting to see how much the area of glass and plastic greenhouses extended in the past 20 years. Which seems to be the main generator of the increased crop production in the area. It seems that irrigation systems brought new development to the area.
Line 365 - "... ramee has been completely replaced.." - ramee?
Line 367: "The proportion of the grain to the total grain output of Guizhou Province increased 367 from 72.55% in 2001 and plummeted to 16.34% in 2020,..." Which grain? Increased or decreased?
Line 373: Figure 6- add the year of the data.
Line 418: Table 7- Explain the abbreviation below the table.
Line 427: "there were only 6340 thousand people" I assume you mean 634,000 people or 0.634 million. Update in the text.
Line 435: Table 8 - what was the population growth? - would that impact the Per capita results?
Line 477-488: The transformation of agriculture in the last 20 years was remarkable. It would be interesting to add how much financial resources were invested by the provincial or central government in the development of the province. I can imagine that that big change in so short time would not be possible without a huge investment in infrastructure and knowledge. This has to be mentioned. I assume five-year plans are an important part of political commitment in China that should not be overlooked in this manuscript. Please add one or two paragraphs on that matter as for now is overlooked in the paper. But it has a significant impact on land sue transformation as well as on the environmental footprint of these investments.
Line 505-533: restructure the text by answering these questions in the text:
Why is this research unique?
What are the shortcomings/uncertainties of this research?
What did the scientific community learn out of it?
What are the benefits/recommendations for stakeholders (farmers, water and environmental managers)?
What are the recommendations for policymakers/legislators?
Future work?
Author Response
We sincerely thank you for the valuable feedback. You will find our point-by-point responses to the comments in the the attachment.

Reviewer 2 Report
In my opinion, the article is generally well written and valuable. In a few places, however, it requires minor corrections / reformulation of the text. My comments concern:
a) the personal mode of sentences that should be changed into impersonal (lines 326, 389, 471, 491, 502, 506),
b) cupboards figure 5 where Evergreen Needleleaf Forest has the same (or a very similar line) to Cropland / Natural Vegetation Mosaisc - this figure can be compared with the content of Table 3 but in my opinion only these 4 categories should be left in the figure (possibly it should appear 5 described as other / other)
c) as for the discussion, there are very few references to the results of other studies - some remarks of this nature were noted in other parts of the text (e.g. the comment about the results of Costanza et al. (lines 277-278) but this part should also be enriched with similar links / content.
Author Response

(The authors gave the same response as above.)
